# Tunneling of Mesh during Ventral Rectopexy: Technical Aspects and Long-Term Functional Results

**DOI:** 10.3390/jcm12010294

**Published:** 2022-12-30

**Authors:** Paola Campennì, Angelo Alessandro Marra, Veronica De Simone, Francesco Litta, Angelo Parello, Carlo Ratto

**Affiliations:** 1Proctology Unit, Fondazione Policlinico Universitario “Agostino Gemelli” IRCCS, 00168 Rome, Italy; 2Department of Medicine and Translational Surgery, Università Cattolica del Sacro Cuore, 00168 Rome, Italy

**Keywords:** ventral rectopexy, rectal prolapse, obstructed defecation syndrome, fecal incontinence, pelvic disorders

## Abstract

Avoiding the extensive damage of pelvic structures during ventral rectopexy could minimize secondary disfunctions. The objective of our observational study is to assess the safety and functional efficacy of a modified ventral rectopexy. In the modified ventral rectopexy, a retroperitoneal tunnel was created along the right side of rectum, connecting two peritoneal mini-incisions at the Douglas pouch and sacral promontory. The proximal edge of a polypropylene mesh, sutured over the ventral rectum, was pulled up through the retroperitoneal tunnel and fixed to the sacral promontory. In all patients, radiopaque clips were placed on the mesh, making it radiographically “visible”. Before surgery and at follow up visits, Altomare, Longo, CCSS, PAC-SYM, and CCFI scores were collected. From March 2010 to September 2021, 117 patients underwent VR. Modified ventral rectopexy was performed in 65 patients, while the standard ventral rectopexy was performed in 52 patients. The open approach was used in 97 cases (55 and 42 patients in modified and standard VR, respectively), while MI surgery was used in 20 cases (10 and 10 patients in modified and standard VR, respectively). A slightly shorter operative time and hospital stay were observed following modified ventral rectopexy (though this was not statistically significant). Similar overall complication rates were registered in the modified vs. standard ventral rectopexies (4.6% vs. 5.8%, *p* = 0.779). At follow-up, the Longo score (14.0 ± 8.6 vs. 11.0 ± 8.2, *p* = 0.042) and “delta” values of Altomare (9.2 ± 6.1 vs. 5.9 ± 6.3, *p* = 0.008) and CCSS (8.4 ± 6.3 vs. 6.1 ± 6.1, *p* = 0.037) scores were significantly improved in the modified ventral rectopexy group. A similar occurrence of symptoms recurrence was diagnosed in the two groups. Radiopaque clips helped to accurately diagnose mesh detachment/dislocation. The proposed modified VR seems to be feasible and safe. Marking the mesh intraoperatively seems useful.

## 1. Introduction

Ventral rectopexy (VR) is an effective surgical option for the treatment of obstructed defecation syndrome (ODS) due to internal rectal prolapse (IRP), enterocele and rectocele, or external rectal prolapse (ERP). D’Hoore et al. [1] first described the VR procedure in patients with ERP, with VR progressively emerging as the procedure of choice for ERP [2,3,4]. To reduce the rectal-prolapse recurrence, several modifications of “standard” VR have been proposed, mainly concerning the mesh placement [5,6,7,8]. However, no superiority has been demonstrated. Since 2014, our group has modified the standard VR by introducing retroperitoneal tunneling of the mesh (preserving both lateral and utero-sacral ligaments) and marking the mesh with radiopaque clips (although radiopaque meshes are already provided), making it “radiologically visible” when mesh detachment or prolapse recurrence should be investigated. The aim of this study was to assess the feasibility and safety of the modified VR and to evaluate its long-term functional outcomes. The results observed in modified VR patients were compared with data from a control group of patients treated with standard VR.

## 2. Materials and Methods

### 2.1. Study Design

The study is reported according to the Strengthening the Reporting of Observational Studies in Epidemiology (STROBE) statement [9]. All patients gave their written, informed consent for data analysis. The study was approved by the ethical committee of the Institution Fondazione Policlinico Universitario Agostino Gemelli IRCCS, Rome, Italy (ID 2574).

Surgery was performed after a complete work-up that included the patient’s history, validated questionnaires administration, a physical examination, and a radiological evaluation (dynamic evacuation proctography or MRI defecography) of the pelvic floor. All patients were discussed by a multidisciplinary pelvic team including proctologists, urogynecologists, and radiologists. All operations were performed by the same expert colorectal surgeon (C.R.).

Demographic data, operation time, hospital stay, and perioperative complications were collected retrospectively and gathered into a database. The recurrence of prolapse, rectocele and/or enterocele, and the functional results were assessed prospectively during the patients’ follow-up visits at 1, 6, and 12 months and then annually.

### 2.2. Patients’ Assessment

Patients’ histories were accurately collected, specifically regarding defecation and urinary disorders, sexual activity impairments, and anatomic abnormalities of pelvic organs. Physical examinations were meticulously performed.

The severity of ODS was assessed by the following validated scoring systems, administered in a face-to-face interview: Altomare score (range: 0–32; no symptoms = 0) [10]; Longo score (range: 0–40; no symptoms = 0) [11]; Cleveland Clinic Constipation Scoring System or CCSS (range: 0–30; no symptoms = 0) [12]; Patient Assessment of Constipation-Symptoms questionnaire or PAC-SYM (including the assessment of twelve items assigned to three subscales, i.e., abdominal, stool, and rectal symptoms; scoring range for each item: 0–4; no symptoms = 0) [13]; and Fecal incontinence (FI) was assessed by the Cleveland Clinic Fecal Incontinence or CCFI score (range: 0–36, perfect continence = 0) [14].

Following the first visit, all patients underwent either a dynamic evacuation proctography or MRI defecography (using intra-rectal, intravaginal, and small bowel contrast, allowing their classification according to the Oxford rectal prolapse grading system) [15], an endoanal ultrasound and anorectal manometry. Colorectal screening was conducted in all cases with a colonoscopy. Indications for surgery were discussed by the multidisciplinary team.

### 2.3. Surgical Technique

Before 2014 (when the proposed new approach with the tunnelling of the mesh was introduced), the traditional technique using the inverted-J-shaped peritoneal incision was used; thereafter, the tunnelling of mesh was used in all patients. Similarly, when the MI approach was introduced in our experience (September 2018), the traditional technique of VR was used along the first period, and then the tunnelling of mesh was performed in all patients.

Preoperatively, bowel preparation (two enemas) and a single dose of broad-spectrum antibiotic were administrated.

Despite using either an open (LT, Pfannenstiel or umbilico-pubic incision) or minimally invasive (MI, robotic, or laparoscopic) approach to the pelvis, the bowel was retracted to the middle-upper abdomen to expose both the pouch of Douglas and the sacral promontory with the patient in the Trendelenburg position.

### 2.4. Modified VR: The Study Group

The modified VR approach provided a 5 cm peritoneal incision at the pouch of Douglas. Rectovaginal space was dissected up to the pelvic floor. The ventral rectum was fully exposed; in case of rectocele, particular attention was paid to the rectovaginal dissection to safely reach the perineum. A combined anorectal–vaginal digital examination confirmed the complete dissection. Any redundant pouch of Douglas was excised so that a longer tract of anterior rectum was available for mesh placement. A strip of polypropylene (Ethicon, Johnson & Johnson, Brussels, Belgium), trimmed to 10–14 cm, was introduced into the abdominal cavity; two radiopaque clips were previously placed on its distal edge. The mesh was sutured over the ventral rectum with a total of six 3-0 PDS sutures arranged. The two most proximal sutures were preserved for further fixation at the proximal vagina. The other two radiopaque clips were placed on mesh sides at the level of its proximal fixation over the rectum.

Thereafter, a small (2–3 cm) peritoneal incision was performed at the level of sacral promontory and the periosteum was freed, avoiding damages to the autonomic nerve fibers (Figure 1). A retroperitoneal tunnel from the promontory incision to the pelvic incision was accurately performed, avoiding damages to lateral and utero-sacral ligaments (Figure 2). The proximal mesh was then pulled up, reaching the sacral promontory. Under a gentle tension (in order to prevent future rectal intussusception and the persistence of rectocele), the mesh was fixed at the sacral promontory with two stitches of 2-0 PDS. Two clips were placed on the mesh and another one was placed on the sacral periosteum to allow for a further radiographic check of the mesh position at follow-up (Figure 3). The redundant mesh was trimmed. To prevent further enterocele, the posterior proximal vagina was approximated to the mesh using the two most proximal sutures left in place before. A full thickness passage with the needle through the posterior vaginal wall should be avoided. The peritoneal incisions were closed with V-loc sutures or absorbable running sutures. A video vignette of the modified VR using robotic approach is available. [16]

### 2.5. Standard VR: The Control Group

A group of patients was treated with the “standard VR,” provided an inverted-J-shaped incision of the pelvic peritoneum [1] and served as the control group. The recto-vaginal dissection, mesh conformation, placement, and fixation at the sacral promontory were all executed similarly to the modified VR.

### 2.6. Postoperative Management

Postoperatively, analgesics drugs were administered. A fiber-enriched diet was generally prescribed and resumed on the second postoperative day. Straining efforts were discouraged.

### 2.7. Data Collection

The mean duration of surgical procedure, intraoperative and postoperative complications (classified with Clavien–Dindo grade system) [17] and their management, conversion rate, postoperative hospital stay, and early reoperations (within 7 days) were analyzed. Follow-ups were scheduled at 1, 6, 12 months, and once a year thereafter. Postoperative symptoms were recorded, as in our daily clinical practice. Altomare, Longo, CCSS, PAC-SYM, and CCFI scores were completed under the supervision of a surgeon at each visit. Functional scores have usually been used in our clinical practice to evaluate and standardize the clinical approach to obstructed defecation syndrome. It is very useful during the first visit, when discussing the clinical case in the multidisciplinary team meeting and to evaluate the results after the treatment. In case of persistent constipation symptoms, patients were deeply investigated using a bowel transit study with radiopaque markers and dynamic evacuation proctography or MRI defecography. If a mesh detachment was suspected, a pelvic X-ray assessment (with antero-posterior, latero-lateral and oblique projections) was performed in order to check the position of the radiopaque clips placed at the operation.

### 2.8. Statistical Analysis

Continuous data were described as means with standard deviations and analyzed by the Wilcoxon and Mann–Whitney tests for paired and unpaired data, respectively. Categorical data were reported as frequencies and percentages and analyzed with a chi-squared test. Clinical scores in modified VR and standard VR patients were compared calculating the differences between preoperative and follow up values (delta-Δ). A *p*-value < 0.05 was considered to be significant. Data were analyzed using IBM SPSS Statistics for Windows, Version 25.0 (IBM Corp, Armonk, NY, USA).

## 3. Results

### 3.1. Demographic Data

From March 2010 to September 2021, 117 patients (116 females) underwent a VR, 38 (32.5%) and 79 (67.5%) for ERP and IRP, respectively. The mean age at the time of operation was 60.8 ± 12.8 years (range: 16 to 94 years). The modified VR was performed in 65 patients (19 patients with ERP, 46 patients with IRP) and the standard VR was performed in 52 patients (19 patients with ERP, 33 patients with IRP).

Further details concerning the clinico-pathological features of patients are reported in Table 1. Concerning patients’ history, 85 patients (73.3%) had at least one vaginal delivery, 66 of them (56.9%) received an episiotomy and 26 (22.4%) reported an obstetric anal sphincter injury; 37 patients (31.9%) were submitted to a hysterectomy. Twenty-three patients (19.7%) were previously submitted to surgery for pelvic floor dysfunctions (Standard VR: 1 Orr-Loygue, 5 STARR, 3 Delorme, 1 Altemeier; Modified VR: 1 Orr-Loygue, 8 STARR, 4 Delorme), with poor outcomes. No difference was detected comparing the two groups.

Regarding clinical features (Table 1), 109 patients (93.2%) had significant impairment of physiologic defecation (including prolonged straining, unsuccessful attempts to defecate, self-digitation for defecation, incomplete evacuation or sensation of anorectal obstruction/blockage, and stool frequency >3 times/week); 72 patients (61.5%) referred symptoms of FI; and 64 patients (54.7%) suffered from both conditions. No statistically significant differences in preoperative clinical scores were reported between the two groups. Urinary incontinence was reported by 69 patients (59.0%), similarly across the two groups.

Patients’ distribution according to the Oxford criteria provided: 11 (9.4%) high recto-rectal intussusceptions (grade 1); 17 (14.5%) low recto-rectal intussusceptions (grade 2); 31 (26.5%) recto-anal intussusceptions onto the anal canal (grade 3), 20 (17.1%) recto-anal intussusceptions into the anal canal (grade 4); and 38 (32.5%) ERPs (grade 5). If stratified in the two groups, this distribution revealed no statistical difference. Interestingly, FI was more frequent in patients with higher Oxford grades (65.2% in grades 4 and 5) in both the modified and standard VR groups (62.2% and 68.6%, respectively; *p* = 0.582).

### 3.2. Operation Data

The modified procedure was introduced in 2014 and it was performed in all patients who gave their written, informed consent. A standard rectopexy was performed in the first period of the study and when the patients denied the consent to the modified procedure.

The open approach was used in 97 cases (55 and 42 patients in modified and standard VR, respectively) while MI surgery was used in 20 cases (10 and 10 patients in modified and standard VR, respectively).

In the modified VR group, six patients (9.2%) needed a conversion to a conventional, inverted-J-shape peritoneal incision; among them, in four LT patients the decision was due to fibrotic tissue secondary to a previous pelvic surgery that made the retroperitoneal tunnelling technically difficult and at risk of damages to the right lateral ligament of the rectum; in two MI cases the conversion was due to bleeding from pelvic varicocele (safely managed intraoperatively) and pelvic adhesions, respectively (Appendix A). During the modified VR, four patients (6.2%) underwent additional surgical procedures: two hemorrhoidectomies, one cystopexy, and one pelvic biopsy. In standard VRs, give patients (9.6%) were submitted to additional procedures: three hemorrhoidectomies, one colposacropexy and one pelvic biopsy. 

Although shorter, no differences in the operative time (92.3 ± 26.8 vs. 94.7 ± 28.9 min, *p* = 0.572, and 133.5 ± 30.7 vs. 145.3 ± 27.2 min, *p* = 0.657, in patients who underwent LT and MI surgery, respectively) and hospital stay (3.1 ± 0.9 vs. 3.3 ± 1.0 days, *p* = 0.114) were observed between the modified and standard VR. An intraoperative complication (bleeding from a pelvic varicocele, controlled by bipolar forceps) was registered in one modified VR patient (0.9%). Five patients (4.3%) experienced postoperative complications: three hematomas, one seroma, one urinary infection (Clavien–Dindo grade ≤ 3). No mesh erosion was recorded. Comparing two groups, the overall complication rates were 4.6% (three patients) in the modified VR group and 5.8% (three patients) in the standard VR group (*p* = 0.779). There were neither re-operations during the primary admission nor perioperative mortalities. No chronic pelvic pain or major morbidities were reported in both VR groups.

### 3.3. Clinical Outcomes

The mean overall follow-up was 40.6 ± 33.1 months (24.3 ± 14.8 and 61.0 ± 38.2 in modified and standard VR, respectively). Following both modified and standard VR, all scores showed a significant reduction compared to baseline (Figure 4).

When comparing the long-term functional scores, the Longo score results were significantly improved after modified VR (*p* = 0.042). The decrease of the mean score from the preoperative to follow-up values (Δ) was always higher in the modified VR than in standard VR patients (Δ-Altomare score: 9.2 ± 6.1 vs. 5.9 ± 6.3, *p* = 0.008; Δ-Longo score: 11.5 ± 8.6 vs. 9.3 ± 9.0, *p* = 0.127; Δ-CCSS: 8.4 ± 6.3 vs. 6.1 ± 6.1, *p* = 0.037; Δ-PAC-SYM: 9.0 ± 9.4 vs. 6.7 ± 7.5, *p* = 0.059; Δ-CCFI score: 2.0 ± 3.6 vs. 1.7 ± 4.2, *p* = 0.900). 

The comparison between pre- and postoperative values of clinical scores in patients with external rectal prolapse and internal rectal prolapse who underwent standard vs. modified ventral rectopexy is shown in Table 2 and Table 3. 

At the last follow up, 20 patients (17.1%) referred persistent constipation symptoms (12 modified VR patients, 18.5%; 8 standard VR patients, 15.4%; *p* = 0.660). All were re-evaluated by clinical and radiological examinations. In 13 cases (11.1%), slow-transit constipation was detected with a satisfying correction of preoperative pelvic organs prolapse; in 5 patients (4.3%), rectocele results were not completely corrected (2 modified VRs and 3 standard VRs). After evaluating the position of radiopaque clips placed intraoperatively with radiological imaging of the pelvis, a mesh detachment from the sacral promontory was diagnosed in 2 patients (1.7%, all standard VRs), probably due to severe and prolonged straining after surgery. These patients underwent surgical mesh re-fixation. Finally, 4 patients (3.4%) had new onset of FI. Preoperative urinary incontinence was reported in 69 patients (59.0%); 39 patients underwent modified VR and 30 patients underwent standard VR (*p* = 0.801). At the last follow-up, postoperative urinary incontinence was observed in 63 patients, 5 of which were cases de novo; 35 in the modified VR group and 28 in the standard VR group (*p* = 1.000).

## 4. Discussion

VR is increasingly favored in the treatment of ERP and has been strongly proposed also for the IRP [18]. Although the literature is not conclusive, the evidence suggests that it is significantly better than posterior rectal prolapse repairs [19]. Its efficacy in regaining the anatomical position of pelvic organs has been largely documented. Moreover, when compared with the procedures using a posterior approach, the VR presents several opportunities: effective management of ERP or IRP; rectocele (even very large), and enterocele (by approximation of posterior vaginal wall to the mesh and ventral rectal wall); and availability for an integrated management of a central or anterior pelvic prolapse (i.e., with colposacropexy).

Although the primary purpose of rectal prolapse surgery would be the correction of the anatomical alterations of the rectum and surrounding structures, the improvement of anatomy does not systematically correspond to improving function due to several factors, some of them occult. The anatomical integrity of fundamental pelvic structures should be preserved, including the entire rectal wall, vascular and nerve supplies (passing through the lateral ligament of the rectum), and pelvic supportive structures, in particular the utero-sacral ligaments. Standard VR (providing a long, inverted-J-shaped peritoneal incision) could be at risk of damage to the right lateral rectal and utero-sacral ligament.

The importance of lateral ligament preservation has been strongly highlighted: they contain autonomic nerves and are fundamental for the rectal motility. Their section/injury could lead to postoperative constipation and dyschezia following surgery (for neoplastic and non-neoplastic rectal diseases), including posterior and postero-lateral rectopexy procedures [20].

Recently, Petros has highlighted the role of utero-sacral ligaments in the biomechanics of pelvic floor organs prolapse [21]. Their pathologic progressive elongation may cause severe uterine prolapse. Moreover, they would progressively splay laterally, causing an enterocele and carrying the lateral rectal wall with them. The wider and weaker anterior rectal wall, due to its lateral stretching [22,23,24], would favor the rectal intussusception. Consequently, the anatomical distortion of the rectum would severely impact its biomechanical properties with significant functional impairments. All these etiopathogenetic events should strongly induce to spare the fundamental pelvic structures during surgery. However, in our opinion, the solution proposed by Petros (i.e., a posterior intravaginal slingplasty operation) would not seem to fit the purpose of an effective management of ERP and IRP. On the other hand, VR has demonstrated to be effective in regaining a correct anatomical organ configuration in the posterior-middle pelvic compartments, leading to significant functional improvements. Unfortunately, the heterogeneity in patient selections and pre- or postoperative assessments makes it impossible to compare the clinical results reported in literature. Among numerous studies published, only one randomized clinical trial compared laparoscopic VR and suture posterior rectopexy for ERP: the former approach gave a lower recurrence rate (8.8% vs. 23.3%) but was not statistically significant (*p* = 0.111), probably due to the small number of studied patients [25]. In a review of 26 studies on laparoscopic VR for ERP and IRP, the improvement of obstructed defecation ranged between 52% and 93%, and that of FI between 48% and 93%. On the contrary, persistent obstructed defecation symptoms may range from 2.6 to 20% [26]. A systematic review and meta-analysis of fourteen studies on laparoscopic VR for IRP reported a weighted complication rate of 13.6% and a mean recurrence of 6.5%. The main causes of recurrence were the incomplete dissection of anterior rectal wall and inadequate mesh fixation and/or position [27]. Later, another systematic review and meta-analysis by Emile et al. [28] on seventeen studies of 1242 patients submitted to a laparoscopic VR for ERP reported a weighted mean improvement of constipation in 71.0% and fecal continence in 79.3% of patients.

The reasons for symptom recurrence are not fully understood, and its predictors are still being investigated [28]. These might include multifactorial pathogenesis of constipation (drugs, depression, anxiety, obesity, etc.), coexistence of a slow-transit bowel constipation, and some technical aspects of surgery regarding the deepness of pelvic dissection, and the mesh chosen and its placement modality, tension, and fixation.

We have perfected the open technique and, more recently, we have also applied it to minimally invasive surgery. Our proposal seeks to improve such good data obtained with standard VR.

Reducing the length of peritoneal incision (splitting the single long peritoneal incision into two small incisions) would decrease the risk of possible iatrogenic damages to both the lateral rectal and utero-sacral ligaments. Although neither prospectively nor randomly performed (significant limitation for this study), the patients’ selection to the operation was homogenous: preoperative patients’ characteristics were similar in both standard and modified VRs in terms of clinico-pathological features and distribution, according to Oxford criteria. The two groups were also similar concerning the preoperative rectal prolapse type (ERP vs. IRP) and the surgical approach chosen (LT vs. MI).

Although other limitations (including the discrepancy of follow-up—which was longer for standard than for modified VRs—and the later adoption of the new technique of mesh tunnelling both in LT and MI groups), the adoption of modified VR was safe, with a low morbidity rate and limited recourse to conversion toward standard “inversed-J” peritoneal incision. The proposed modified VR obtained a slight reduction (not statistically significant) in operative time and postoperative hospital stay. A larger experience could further improve such results.

Concerning both ODS and FI, both the standard and modified VRs showed significant improvements in the severity of patients’ symptoms. The modified VR produced a significantly lower Longo score (*p* = 0.042), and higher “Δ” of Altomare and CCSS scores (*p* = 0.008 and *p* = 0.037, respectively) between baseline and follow-up values, confirming a good trend of modified VR. When comparing the functional results for the two proposed rectopexy techniques, both techniques seem to be effective. This does not allow us to demonstrate a certain advantage of the modified approach, but certainly it seems to be a surgical option to be considered and studied further.

Interestingly, the adoption of marking the mesh with a few radiopaque clips during the operation allowed for its identification in follow-up. When a mesh detachment/dislocation was suspected, a simple pelvic X-ray series addressed the repair when confirmed, or to avoid a useless surgery when denied.

## 5. Conclusions

The retroperitoneal tunneling of mesh during the proposed, modified VR appears to be feasible and safe. Its long-term clinical results seem to be promising but without a great functional improvement. Further large and multicentric randomized trials will verify and clarify its role in managing ERP and IRP. Suspicious mesh detachment/dislocation can be settled simply and easily radiographically checking the position of intraoperatively placed radiopaque clips.

## Figures and Tables

**Figure 1 jcm-12-00294-f001:**
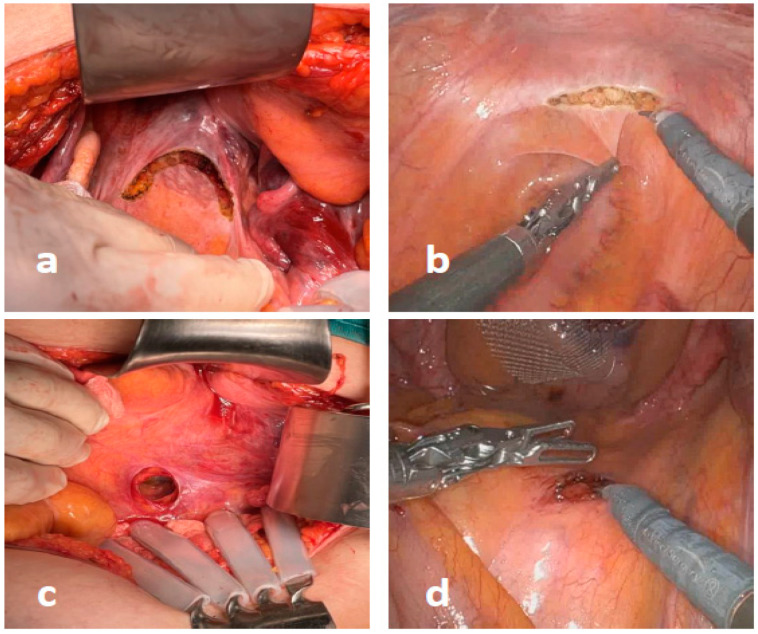
Peritoneal incisions were performed at the pouch of Douglas (**a**,**b**) and sacral promontory (**c**,**d**) in open and robotic surgery.

**Figure 2 jcm-12-00294-f002:**
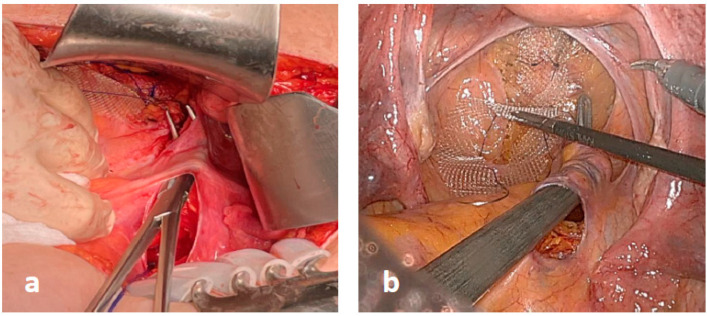
A retroperitoneal tunnel was created in open (**a**) and robotic (**b**) surgery.

**Figure 3 jcm-12-00294-f003:**
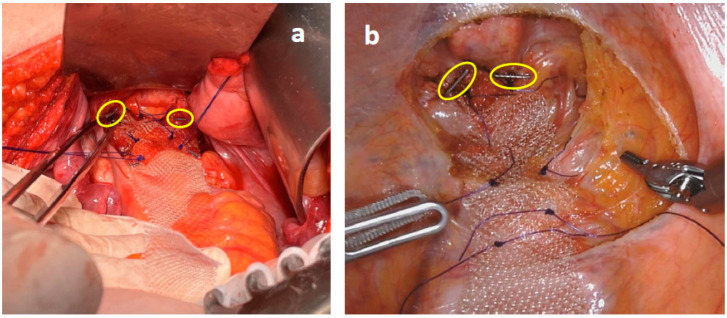
The mesh was sutured over the ventral rectum with three couples of 3-0 PDS stitches in open (**a**) and robotic (**b**) surgery. Two radiopaque clips were positioned at the distal edge of the mesh (circles).

**Figure 4 jcm-12-00294-f004:**
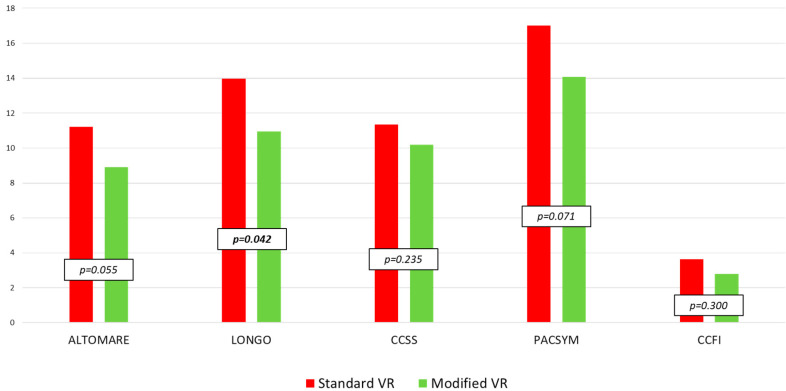
Comparison between postoperative values of clinical scores in patients who underwent standard vs. modified ventral rectopexy (Mann–Whitney U test). Altomare = Altomare score; Longo = Longo score; CCSS = Cleveland Clinic Constipation Scoring System; PACSYM = Patient Assessment of Constipation-Symptoms questionnaire; CCFI = Cleveland Clinic Fecal Incontinence score.

**Table 1 jcm-12-00294-t001:** Baseline characteristics of patients who underwent modified and standard ventral rectopexy.

	Modified VR—n° (%)	Standard VR—n° (%)	*p*
**Patients**	65	52	
**Female**	65 (100.0)	51 (98.1)	*0.262*
**Age (years) ***	60.9 ± 12.0	60.7 ± 13.9	*0.251*
**Previous abdominal surgery**	39 (60.0)	32 (61.5)	*0.866*
** Previous hysterectomy**	19 (29.2)	18 (35.3)	*0.487*
** Previous abdominal surgery for RP**	1 (1.5)	1 (1.9)	*0.873*
**Previous perineal surgery**	26 (40.0)	21 (40.4)	*0.966*
** Previous perineal surgery for RP**	12 (18.5)	9 (17.3)	*0.645*
**Vaginal delivery**	45 (69.2)	40 (78.4)	*0.266*
** Episiotomy**	35 (53.8)	31 (60.8)	*0.454*
** Obstetric anal sphincter injury**	9 (13.8)	17 (33.3)	*0.188*
** Forceps/vacuum**	1 (1.5)	6 (11.8)	*0.402*
**Caesarean delivery**	13 (20.0)	10 (19.6)	*0.958*
**Preoperative ODS**	61 (93.8)	48 (92.3)	*0.743*
**Preoperative FI**	37 (56.9)	35 (67.3)	*0.251*
**Preoperative ODS + FI**	33 (50.8)	31 (59.6)	*0.339*
**Preoperative UI**	39 (60.0)	30 (57.7)	*0.801*
**Oxford classification for RP**			*0.508*
** Grade I**	7 (10.8)	4 (7.7)	
** Grade II**	12 (18.5)	5 (9.6)	
** Grade III**	18 (27.7)	13 (25.0)	
** Grade IV**	9 (13.8)	11 (21.2)	
** Grade V**	19 (29.2)	19 (36.5)	
**Rectocele**	51 (78.5)	30 (57.7)	*0.174*
**Enterocele**	35 (53.8)	35 (67.3)	*0.677*

* Data are shown as mean ± standard deviation. VR = ventral rectopexy; RP = rectal prolapse; ODS = obstructed defecation syndrome; FI = fecal incontinence; UI = urinary incontinence.

**Table 2 jcm-12-00294-t002:** Comparison between pre- and postoperative values of clinical scores in patients with only internal rectal prolapse who underwent standard vs. modified ventral rectopexy (Mann–Whitney U test).

	Standard VR	SD	Modified VR	SD	*p*
**Altomare pre**	17.8	5.6	17.8	5.8	*0.952*
**Longo pre**	23.7	7.5	22.0	8.8	*0.489*
**CCSS pre**	18.2	4.9	18.7	5.4	*0.383*
**PAC-SYM pre**	23.9	5.5	23.2	5.6	*0.643*
**CCFI pre**	4.5	5.0	3.2	4.7	*0.176*
**Altomare post**	10.8	5.9	8.7	6.5	*0.116*
**Longo post**	13.5	7.8	10.5	8.4	*0.050*
**CCSS post**	11.4	5.4	10.0	6.3	*0.289*
**PAC-SYM post**	16.0	7.7	13.8	10.0	*0.155*
**CCFI post**	3.0	4.3	2.2	3.7	*0.211*

VR = ventral rectopexy; SD = standard deviation; Pre = preoperative; Post = postoperative; Altomare = Altomare score; Longo = Longo score; CCSS = Cleveland Clinic Constipation Scoring System; PAC-SYM = Patient Assessment of Constipation-Symptoms questionnaire; CCFI = Cleveland Clinic Fecal Incontinence score.

**Table 3 jcm-12-00294-t003:** Comparison between pre- and postoperative values of clinical scores in patients with only external rectal prolapse who underwent standard vs. modified ventral rectopexy (Mann–Whitney U test).

	Standard VR	SD	Modified VR	SD	*p*
**Altomare pre**	15.9	6.6	18.6	6.6	*0.242*
**Longo pre**	22.1	7.7	24.4	7.7	*0.368*
**CCSS pre**	16.3	6.2	19.0	4.6	*0.106*
**PAC-SYM pre**	23.8	7.4	23.7	5.8	*0.927*
**CCFI pre**	6.8	6.8	8.6	5.5	*0.278*
**Altomare post**	11.9	7.2	9.4	6.3	*0.361*
**Longo post**	14.9	10.2	12.2	7.6	*0.447*
**CCSS post**	11.2	6.4	10.5	5.8	*0.703*
**PAC-SYM post**	18.9	9.8	14.8	9.3	*0.207*
**CCFI post**	4.8	5.7	4.3	4.6	*0.951*

VR = ventral rectopexy; SD = standard deviation; Pre = preoperative; Post = postoperative; Altomare = Altomare score; Longo = Longo score; CCSS = Cleveland Clinic Constipation Scoring System; PAC-SYM = Patient Assessment of Constipation-Symptoms questionnaire; CCFI = Cleveland Clinic Fecal Incontinence score.

## Data Availability

The data presented in this study are available on request from the corresponding author.

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
