# Peer review of "Tunneling of Mesh during Ventral Rectopexy: Technical Aspects and Long-Term Functional Results"

_jcm, 2022, doi:10.3390/jcm12010294_

Round 1

Reviewer 1 Report

Since the study is from 3/2010 to 9/2021, and the authors modified the standard VR since 2014, is this study 52 standard VR from 2010-2013 and 65  modified VR from 2014-2021, or some overlap? If YES, this is a historical control trial, and how can the authors not be sure any results are due to learning curve and improved technique?

If NO, why do the authors state they modified their technique since 2014 and have not had all surgeons change? What was the criteria or surgeon preference for one procedure vs the other.

Clearer presentation of methods and timing and numbers of standard and modified VR must be included to understand what they are comparing.

Laparotomic is not a word in my dictionary. I understand it refers to surgery via laparotomy, but would be best called 'Open'.

Why the wide variance in approach? Open and MIS techniques.Does this make interpretation harder?

Umbilico-pubic NOT Umbelico-pubic

Discussion: Suggest start with VR is increasingly favored in the treatment...

Author Response

Since the study is from 3/2010 to 9/2021, and the authors modified the standard VR since 2014, is this study 52 standard VR from 2010-2013 and 65  modified VR from 2014-2021, or some overlap? If YES, this is a historical control trial, and how can the authors not be sure any results are due to learning curve and improved technique?

If NO, why do the authors state they modified their technique since 2014 and have not had all surgeons change? What was the criteria or surgeon preference for one procedure vs the other.

Clearer presentation of methods and timing and numbers of standard and modified VR must be included to understand what they are comparing.

Thanks, we introduced the modified technique from 2014 and proposed it to all our patients. To be clear, we have added a specification in the Methods section and a table with a sort of algorithm as supplementary material.

The learning curve to create a retroperitoneal tunnel was very short, considering that it is an easy and fast surgical procedure. In section “Operation data” we reported the number of modified procedures in open and MI approaches and when and why we converted from modified to standard rectopexy.

Laparotomic is not a word in my dictionary. I understand it refers to surgery via laparotomy, but would be best called ‘Open'.

Thanks, we corrected the text.

Why the wide variance in approach? Open and MIS techniques.Does this make interpretation harder?

We performed only 20 MI VR because we performed the MI approach only in the last 3 years of the study.

Umbilico-pubic NOT Umbelico-pubic

Thanks, we corrected the text.

Discussion: Suggest start with VR is increasingly favored in the treatment…

Thanks, we corrected the text.

Reviewer 2 Report

This is a case - control study of patients with pelvic organ prolapse that have been operated on by two different ventral rectopexy surgical approaches, conventional ventral rectopexy and the modified version proposed by authors.

Although quite interesting as it suggests a new technical alternative, some issues can be improved previous to manuscript acceptance:

- It can be considered more a case-control study than a cohort study as you are comparing a new approach (modified ventral rectopexy / cases) with those patients with the standard approach (conventional ventral rectopexy / controls).

- It should be stated how all the functional tests that are stated were recorded in a retrospective way. 

- The composition of the multidisciplinary team would be quite illustrative in the material and methods section.

- Statistical analysis, or at least comparisons between figures 3 and 4 are hardly understandable. What statistics were applied to compare the results obtained in both groups and to justify superiority of the new approach?

- An additional figure comparing peritoneal incisions done in both techniques could also help to understand the differences between the two techniques. In the description, and also in the figures, a 5 cm incision over the Douglas´ pouch and an extra one at the promontory looks like quite similar to the conventional J-shape peritoneal incision. Apart from that, avoiding damages to vascular and nervous structures can be achievable both ways.

- Follow-up times are different in both groups. Considering that a functional disease is being analyzed, is it reasonable to expect a progressive functional deterioration of the modified ventral rectopexy group?

Author Response

This is a case - control study of patients with pelvic organ prolapse that have been operated on by two different ventral rectopexy surgical approaches, conventional ventral rectopexy and the modified version proposed by authors.

Although quite interesting as it suggests a new technical alternative, some issues can be improved previous to manuscript acceptance:

  • It can be considered more a case-control study than a cohort study as you are comparing a new approach (modified ventral rectopexy / cases) with those patients with the standard approach (conventional ventral rectopexy / controls).

Thanks, we corrected the text.

  • It should be stated how all the functional tests that are stated were recorded in a retrospective way. 

Thanks, we corrected the text. Postoperative symptoms were recorded, as in our daily clinical practice. Altomare, Longo, CCSS, PAC-SYM, and CCFI scores were completed under the supervision of a surgeon at each visit. Functional data were analyzed retrospectively.

  • The composition of the multidisciplinary team would be quite illustrative in the material and methods section.

Thanks, we did it.

  • Statistical analysis, or at least comparisons between figures 3 and 4 are hardly understandable. What statistics were applied to compare the results obtained in both groups and to justify superiority of the new approach?

  • Thanks, we modified the figure, Comparison between pre- and postoperative values of clinical scores in patients who underwent standard vs modified ventral rectopexy (Mann-Whitney U test).

  • An additional figure comparing peritoneal incisions done in both techniques could also help to understand the differences between the two techniques. In the description, and also in the figures, a 5 cm incision over the Douglas´ pouch and an extra one at the promontory looks like quite similar to the conventional J-shape peritoneal incision. Apart from that, avoiding damages to vascular and nervous structures can be achievable both ways.

Thanks, we have added a picture showing the proposed double incisions at Douglas and sacral promontory. As you can see, the proposed approach is significantly different than the traditional inverted-J incision. In particular, please note that both the right lateral ligament of rectum and the right uterosacral ligament are not at risk of lesion.

  • Follow-up times are different in both groups. Considering that a functional disease is being analyzed, is it reasonable to expect a progressive functional deterioration of the modified ventral rectopexy group?

Thanks for the comment. The shorter follow-up in the modified group (24.3±14.8 and 61.0±38.2 in modified and standard VR, respectively) is a declared limit of the study regarding the long-term functional results. However, the follow up period in the modified VR group seems long enough to be considered a long-time follow up. In any comparison between an old technique vs. a new technique, analyzed retrospectively, you would find the same issue. Future prospective studies are mandatory to investigate it.

Reviewer 3 Report

This is an interesting retrospective cohort study comparing two techniques for ventral rectopexy in patients with internal and external rectal prolapse with an acceptable long period for follow-up. The authors claim that their technique of retroperitoneal tunneling for the mesh leads to better functional results by preserving the lateral ligaments. Furthermore, marking the mesh with radiopaque clips is supposed to be helpfull in case of recurrent symptoms in order to rule out mesh dislocation, as occured in 2 patients.

Though this is an intersting study some major objections must be raised:

The conclusion „some apparent improvements in the long-term functional results“ (line 24) is not supported by the data presented.

Abstract: Please give numbers and p-values for operative time and hospital stay, specify complications for both groups. Give numbers and p-values for Longo, Altomare, and CCSS Scores. Readers would then recognize that there may be a trend towards improvements, however, these differences aren’t statistically significant and therefore not proven by the present study.

It must be made recognizable clearly to the readers that the vast majority of the procedures in both groups were performed by laparotomy (83%!), wich might need explanation in times of minimally invasive surgery.

What was the criteria for using the modified technique? It appears that the modified technique was compared to a historical cohort from a period before the introduction of the modification, please explain.

Please differenciate results for internal and external recal prolaps, since symptoms differ.

 Figures 3 and 4 must be revised in order to compare both techniques directly, making it visible that most items were not statistically different.

Minor objections:

Line 41: „results were matched“ What were the matching criteria? To my understanding, there was no matching performed, but consecutive patients from different periods were compared.

Methods: Line 86: Please specify the approaches for LT, Pfannenstiel, umbilico-pubic incisions or robotic and laparoscopic in both groups. Is this really comparable?

 Why do the authors use resorbable sutures for fixation to the promontory and then wonder about mesh dislocation at this site?

 Marking with clips: radiopaque meshes are already provided by several companies, please mention.

 There is a remarkably high percentage (19.7%) of revisions after previous pelvic floor surgery (line 178). How was the distribution for each procedure among bothe groups? If the lateral ligaments were already divided in prior surgery, how should the modified technique still be favorable? Please explain. Were intra- and postoperative results impaired due to previous surgery?

 Did urinary symptoms improve in either group?

 Line 221: How were postoperative complications distributed among the groups?

 Lines 233-236: „The decrease was higher….“ Please differentiate between statistically significant results and trends!

 Line 240: Slow transit in 11% as a reason for persisting symptoms. This must had been ruled out preoperatively since the ventral rectopexy is not the operation of choice in such cases.

 Line 274 „some occult“ better „some of them occult“?

Line 264 „although insufficient is the evidence“ Language editing required

 Line 330: again, not significant, only a trend

Author Response

This is an interesting retrospective cohort study comparing two techniques for ventral rectopexy in patients with internal and external rectal prolapse with an acceptable long period for follow-up. The authors claim that their technique of retroperitoneal tunneling for the mesh leads to better functional results by preserving the lateral ligaments. Furthermore, marking the mesh with radiopaque clips is supposed to be helpfull in case of recurrent symptoms in order to rule out mesh dislocation, as occured in 2 patients.

Though this is an intersting study some major objections must be raised:

The conclusion „some apparent improvements in the long-term functional results“ (line 24) is not supported by the data presented.

Thanks for the comment. We removed that sentence.

Abstract: Please give numbers and p-values for operative time and hospital stay, specify complications for both groups. Give numbers and p-values for Longo, Altomare, and CCSS Scores. Readers would then recognize that there may be a trend towards improvements, however, these differences aren’t statistically significant and therefore not proven by the present study.

Thanks for the comment. We did it.

It must be made recognizable clearly to the readers that the vast majority of the procedures in both groups were performed by laparotomy (83%!), wich might need explanation in times of minimally invasive surgery.

Thanks for the comment. In our historical experience, the approach to VR was traditionally via open laparotomy, and, with the introduction of “tunnelling” technique, both the traditional inverted-J approach and the new tunnelling approach were used. Similarly, when the minimally invasive surgery has been adopted, both techniques were used. We think that the approach used (open vs. minimally invasive) did not change the specific different constructions of the VR (inverted-J vs. tunnelling). We have specified that in the text and with a figure.

What was the criteria for using the modified technique? It appears that the modified technique was compared to a historical cohort from a period before the introduction of the modification, please explain.

Thanks for the comment. In both the LT group and MI group, standard rectopexy was performed in the first period. Then, since 2014, the modified procedure was introduced and performed in all patients.

Please differenciate results for internal and external recal prolaps, since symptoms differ.

Thanks for the comment. We added two tables to show the Comparison between pre- and postoperative values of clinical scores in patients with external rectal prolapse and internal rectal prolapse who underwent standard vs modified ventral rectopexy.

 Figures 3 and 4 must be revised in order to compare both techniques directly, making it visible that most items were not statistically different.

Thanks for the comment. We modified the figures.

Minor objections:

Line 41: „results were matched“ What were the matching criteria? To my understanding, there was no matching performed, but consecutive patients from different periods were compared.

Thanks for the comment. We modified it.

Methods: Line 86: Please specify the approaches for LT, Pfannenstiel, umbilico-pubic incisions or robotic and laparoscopic in both groups. Is this really comparable?

Thanks for the comment. We reported in results the numbers of both groups. We don’t compared the technical approach to the pelvis (Open vs MI), but we considered it to evaluate the impact of modified VR in the same group.

 Why do the authors use resorbable sutures for fixation to the promontory and then wonder about mesh dislocation at this site?

Thanks for the comment. We used PDS sutures as suggested in the: Consensus on ventral rectopexy: report of a panel of experts.

Mercer-Jones MA, D’Hoore A, Dixon AR et al _Colorectal Dis 2014;16:82-8

 Marking with clips: radiopaque meshes are already provided by several companies, please mention.

Thanks for the comment. Because in our experience, the mesh is tailored on the base of prolapse characteristics, the use of the meshes you mention could not guarantee a reliable imaging when necessary. Our proposal is simple, easy, safe and adaptable to any condition.

 There is a remarkably high percentage (19.7%) of revisions after previous pelvic floor surgery (line 178). How was the distribution for each procedure among bothe groups? If the lateral ligaments were already divided in prior surgery, how should the modified technique still be favorable? Please explain. Were intra- and postoperative results impaired due to previous surgery?

Thanks for the comment. We added in the text the distribution of previous pelvic floor surgery per groups,

The intraoperative impact of previous surgery was reported in section operative data. The postoperative impact of previous surgery was reported in results.

 Did urinary symptoms improve in either group?

Thanks for the comment. We added it in the text. No significant difference was reported.

 Line 221: How were postoperative complications distributed among the groups?

 Thanks for the comment. We reported in operation data: Similar overall complication rates were registered in modified vs. standard ventral rectopexy (4.6% - 1 bleeding, 1 hematoma and 1 urinary infection - vs. 5.8% - 1 seroma and 2 hematomas, p=0.779).

 Lines 233-236: „The decrease was higher….“ Please differentiate between statistically significant results and trends!

 Thanks for the comment, we reported the p-value to underline the statistically significant results

 Line 240: Slow transit in 11% as a reason for persisting symptoms. This must had been ruled out preoperatively since the ventral rectopexy is not the operation of choice in such cases.

Thanks for the comment. In these cases the slow transit was contemporaneous to the rectal prolapse.

 Line 274 „some occult“ better „some of them occult“?

 Thanks for the comment. We changed it.

Line 264 „although insufficient is the evidence“ Language editing required

Thanks for the comment. We changed it.

 Line 330: again, not significant, only a trend

Thanks for the comment. We changed it.

Round 2

Reviewer 2 Report

Supplementary material 1 has to be improved in editing as in the present form is incomprehensible.

Information given in tables 2 and 3 overlaps with information given in figure 4 and, in fact, statistical significance is different in both of them, probably because tables 2 and 3 presents the information separately for internal rectal prolapse and external rectal prolapse, and figure 4 presents the results all together. Despite this, there are just two results that can be considered statistically significant along all of them and, no mention is done regarding what can be considered clinically relevant differences for each of the tests included. For example, the score of Longo in the postoperative global Cohort is 14 in the new group vs. 11 in the classical group, but both of them were quite higher preoperatively, is that 3 points clinically relevant in the Longo score, or they should not be considered as such and the only important message is that patients experienced a great improvement with any of the techniques used?

You used several scores in your evaluation of patients but it cannot be interpreted from the manuscript the real value in clinical terms that all these tests translate after your investigation. 

I consider this a crucial aspect of the paper that should be improved in order to make it much more atractive. The discussion just dedicated a single paragraph to discuss functional results: "Both standard and modified VRs showed significant improvements of patients’ symptoms severity, concerning both ODS and FI. Modified VR produced a significantly lower Longo score (p=0.042), and higher “Δ” of Altomare and CCSS scores (p=0.008 and p=0.037 respectively) between baseline and follow up values, confirming the good trend of modified VR". In addition, the p values of 0.008 and 0.037 mentioned in the discussion section could not be found along the results. Please explain.

In addition, conclusions should also be soften as the differences among the classical and the new approach have not demonstrated to achieve such a different clinical outcome.

Author Response

Supplementary material 1 has to be improved in editing as in the present form is incomprehensible.

Thanks, we had modified it

Information given in tables 2 and 3 overlaps with information given in figure 4 and, in fact, statistical significance is different in both of them, probably because tables 2 and 3 presents the information separately for internal rectal prolapse and external rectal prolapse, and figure 4 presents the results all together. Despite this, there are just two results that can be considered statistically significant along all of them and, no mention is done regarding what can be considered clinically relevant differences for each of the tests included. For example, the score of Longo in the postoperative global Cohort is 14 in the new group vs. 11 in the classical group, but both of them were quite higher preoperatively, is that 3 points clinically relevant in the Longo score, or they should not be considered as such and the only important message is that patients experienced a great improvement with any of the techniques used?

Thanks, we had modified the figure 4 in order to improve the presentation and interpretation of the functional results.

We added in the disccussion: Compering the functional results for the two proposed rectopexy techniques, both technique seems to be effective. A single score (Longo score) is noted with statistically significant results in favor of tunneling. This does not allow us to demonstrate a certain advantage of the modified approach, but certainly it seems to be a surgical option to be considered and studied further. Prospective randomized trials could clarify any doubts.

You used several scores in your evaluation of patients but it cannot be interpreted from the manuscript the real value in clinical terms that all these tests translate after your investigation. 

As reported in line 163, these functional scores have usually been used in our clinical practice to evaluate and standardize the clinical approach to obstructed defecation syndrome. It is very useful during the first visit, when discussing the clinical case in the multidisciplinary team meeting and to evaluate the results after the treatment.

I consider this a crucial aspect of the paper that should be improved in order to make it much more atractive. The discussion just dedicated a single paragraph to discuss functional results: "Both standard and modified VRs showed significant improvements of patients’ symptoms severity, concerning both ODS and FI. Modified VR produced a significantly lower Longo score (p=0.042), and higher “Δ” of Altomare and CCSS scores (p=0.008 and p=0.037 respectively) between baseline and follow up values, confirming the good trend of modified VR". In addition, the p values of 0.008 and 0.037 mentioned in the discussion section could not be found along the results. Please explain.

Thanks in the results in line 258 were reported these p values.

In addition, conclusions should also be soften as the differences among the classical and the new approach have not demonstrated to achieve such a different clinical outcome.

Thanks, we had modified it.

Reviewer 3 Report

The revised manuscript contains some improvements, but still, with respect, the authors don't differentiate properly between trends and statistically significant differences. This is a major flaw:

There are no statistically significant differences in operation time and hospital stay between the groups. This is not stated properly in the abstract or in the discussion part, although authors claim to have done so in their rebuttal letter.

Among the wide variety of scores, most scores were not significantly different between the two groups. The authors seem to avoid this clear statement and it appears like this can only be recognized after careful reading of text and tables. Only the Longo-score seems to have better postoperative results, whereas differences in deltas (Altomare and CCSS) rather appear to serve as surrogate markers as the scores themselves don't offer significant differences between the groups. Please look at the tables 3 and 4: no statistically differences in postoperative results in any parameter. Also the modified figure 4 doesn't give a clear comparision, it is not clear to which items the p-values refer to and the figure legend doesn't give a proper explanation to what is demonstrated. 

Please indicate clearly in the abstract that the vast majority of the operations was performed in an open fashion. This isn't a shame, it is good surgery.

Please don't sugarcoat your results, this isn't neccessary as your results still demonstrate that the modified technique is adequate for both open and minimally invasive rectopexy, which is worth publishing. But obviously, the modified technique is not superior, or at least it's not shown by your data unequivocally. As I said, the manuscript is interesting, but a more modest presentation would have been preferable. 

Author Response

The revised manuscript contains some improvements, but still, with respect, the authors don't differentiate properly between trends and statistically significant differences. This is a major flaw:

There are no statistically significant differences in operation time and hospital stay between the groups. This is not stated properly in the abstract or in the discussion part, although authors claim to have done so in their rebuttal letter.

Results Line 243

no differences in the operative time (92.3±26.8 vs. 94.7±28.9 minutes, p=0.572, and 133.5±30.7 vs. 145.3±27.2 minutes, p=0.657, in patients who underwent LT and MI surgery, respectively) and hospital stay (3.08±0.9 vs. 3.3±1.0 days, p=0.114) were observed between modified and standard VR

Discussion Line 375

The proposed modified VR obtained a slight reduction of operative time and postoperative hospital stay: a larger experience could furtherly improve such results  We added not statistically significant in the text.

Among the wide variety of scores, most scores were not significantly different between the two groups. The authors seem to avoid this clear statement and it appears like this can only be recognized after careful reading of text and tables. Only the Longo-score seems to have better postoperative results, whereas differences in deltas (Altomare and CCSS) rather appear to serve as surrogate markers as the scores themselves don't offer significant differences between the groups. Please look at the tables 3 and 4: no statistically differences in postoperative results in any parameter. Also the modified figure 4 doesn't give a clear comparision, it is not clear to which items the p-values refer to and the figure legend doesn't give a proper explanation to what is demonstrated. 

Thanks, we modified the text and the figure 4.

Please indicate clearly in the abstract that the vast majority of the operations was performed in an open fashion. This isn't a shame, it is good surgery.

Thanks, we added it.

Please don't sugarcoat your results, this isn't neccessary as your results still demonstrate that the modified technique is adequate for both open and minimally invasive rectopexy, which is worth publishing. But obviously, the modified technique is not superior, or at least it's not shown by your data unequivocally. As I said, the manuscript is interesting, but a more modest presentation would have been preferable.

Thanks, we underlined the not superiority of modified VR in the discussion and conclusion.